# The Impact of Common Recovery Blood Sampling Methods, in Mice (Mus Musculus), on Well-Being and Sample Quality: A Systematic Review

**DOI:** 10.3390/ani10060989

**Published:** 2020-06-05

**Authors:** Alexandra L Whittaker, Timothy H Barker

**Affiliations:** 1School of Animal and Veterinary Sciences, The University of Adelaide, Roseworthy Campus, Roseworthy, South Australia 5371, Australia; 2JBI, Faculty of Health and Medical Sciences, The University of Adelaide, South Australia 5005, Australia; timothy.barker@adelaide.edu.au

**Keywords:** mouse, blood sample, well-being, retrobulbar, submandibular, sublingual

## Abstract

**Simple Summary:**

Blood sampling is often performed in laboratory mice. Whilst the techniques are likely to cause only momentary pain or distress, given their frequency of performance, it is essential that the method which best safeguards welfare is used. The small size of mice makes sampling challenging, and use of some routes is controversial due to perceived impact on animal welfare. However, to date, no summary of the evidence relating to welfare impacts arising from these techniques has been presented. This paper presents a systematic review of the literature, with quality appraisal of the studies and an assignment of certainty in the evidence. We conclude that there is not enough high-quality evidence available to make a determination on optimal blood sampling route. We provide recommendations for improving future laboratory animal welfare research through standardisation of outcome measures and enhanced adherence to experimental design and reporting guidelines.

**Abstract:**

Blood sampling is often performed in laboratory mice. Sampling techniques have the potential to cause pain, distress and impact on lifetime cumulative experience. In spite of institutions commonly providing guidance to researchers on these methods, and the existence of published guidelines, no systematic evaluation of the evidence on this topic exists. A systematic search of Medline, Scopus, and Web of Science was performed, identifying 27 studies on the impact of recovery blood sample techniques on mouse welfare and sample quality. Studies were appraised for quality using the SYstematic Review Centre for Laboratory animal Experimentation (SYRCLE) risk of bias tool. In spite of an acceptable number of studies being located, few studies examined the same pairwise comparisons. Additionally, there was considerable heterogeneity in study design and outcomes, with many studies being at a high risk of bias. Consequently, results were synthesised using the Synthesis Without Meta-analysis (SWiM) reporting guidelines. Grading of Recommendations, Assessment, Development and Evaluation (GRADE) was utilised for assessment of certainty in the evidence. Due to the heterogeneity and GRADE findings, it was concluded that there was not enough high-quality evidence to make any recommendations on the optimal method of blood sampling. Future high-quality studies, with standardised outcome measures and large sample sizes, are required.

## 1. Introduction

It is common in biomedical research for protocols to require blood collection from mice (Mus musculus) in order to measure a range of circulating products. The small size of these animals makes such procedures challenging, but a range of sampling methods are documented and widely used. Common considerations in the selection of sample techniques are their practicality and ease of use, the ability to attain the desired blood volume, sample quality and impact on animal well-being [1,2]. 

Retrobulbar bleeding (retroorbital) enables acquisition of larger blood volumes (e.g., 0.2–1 mL) [3], but has been controversial due to the risk of substantial tissue damage to the eye [3,4]. Anecdotally, it appears that this technique has fallen out of favour, particularly in some jurisdictions, such as Australia. This has led to the development of alternative methods. The most common alternative is facial (also commonly called submandibular) vein venepuncture [5]. Despite the rise in popularity of this method, perhaps driven by the aesthetically repugnant use of the ocular area, veins in the facial region lie beneath other important tissues such as glandular tissue [6]. This method then also poses a risk of causing secondary complications via tissue damage. Sublingual vein puncture is another alternative method which yields large-volume collections [4]. In contrast to the submandibular technique, sublingual sampling is generally performed under anaesthesia to immobilise the animal [4]. Anaesthesia, as an adjunct, has the potential to impact both positively and negatively on animal well-being, through minimisation of tissue damage [1], or ‘hangover’ effects from the drugs. Anaesthetic use will also influence the practicality of the technique due to equipment needed and the time taken to perform. 

A range of methods are available for the attainment of smaller volume samples (approximately 0.1–0.15 mL), or for frequent repeat sampling. The tail is commonly used as a blood collection site. A range of collection techniques are described, including targeted collection from the lateral tail vein [7,8], tail tip amputation [9,10,11], and tail incision through cut of the veins [12,13]. Anaesthesia has been regarded as unnecessary for this site, enabling multiple repeat samples. However, warming of the animal may be required to encourage vasodilation [3]. This may add to overall impact on animal well-being. The lateral tarsal or saphenous vein is a common alternative to the tail. Similarly, it requires no anaesthetic and has the added advantage of not requiring external methods for vasodilation. Removal of the scab enables serial blood sampling [3].

Whilst pain, discomfort and physiological stress arising from blood sampling are likely to be short-lived, as one of the most common procedures performed on laboratory animals, researchers and animal ethics committees have a duty to utilise or promote the method with least impact on animal well-being. Furthermore, with the demise of retrobulbar sampling on supposed ethical grounds, it is imperative that an evidence-based approach to the selection of alternative methods is used. Whilst there have been a range of studies investigating the impact of a number of the sampling techniques on mice well-being, these studies typically only contrast a few techniques, and are practically limited in terms of sample sizes. The aim of this systematic review is to present the evidence related to common recovery blood sampling techniques in mice, with regard to animal well-being. Through identification of all relevant evidence, assimilation of study findings to increase statistical power, and study appraisal, it is our intention that this systematic review will provide increased strength of evidence to better inform researchers, ethics committees, and policy makers in their decision making. 

## 2. Materials and Methods 

A priori protocol was created for this review and has been registered on the SYstematic Review Centre for Laboratory animal Experimentation (SYRCLE) database for animal intervention studies [14]. 

### 2.1. Eligibility Criteria

Inclusive criteria were as follows: (P) studies that include post-weaning inbred or outbred laboratory mice. Neonatal/pre-weaning mice were excluded; (I) studies that evaluated recovery, non-surgical blood sampling techniques. Included techniques were: sublingual, retrobulbar sinus, facial, tail sampling methods, and saphenous vein. Studies that examined both one-off and serial sampling were eligible for inclusion; (C) studies were included that compared the intervention to no blood sample, or other included recovery sample method. Studies with no control group (observational studies) were also eligible for inclusion; (O) outcomes such as mortality, quantifiable measures of mouse well-being such as behavior change, bodyweight change, morbidity and quantifiable measures of sample quality such as hemolysis were included. Outcomes in either the immediate post-sampling period or over the longer term were considered for inclusion; and (S) experimental and quasi-experimental study designs including randomised controlled trials, non-randomised controlled trials, and before and after studies were eligible for inclusion. Observational studies were considered for inclusion. 

### 2.2. Search Strategy

The search strategy aimed to locate published studies in English. An initial limited search of Medline was undertaken to identify articles on the topic. The text words contained in the titles and abstracts of relevant articles, and the index terms used to describe the articles were used to develop a full search strategy for Medline. The search strategy, including all identified keywords and index terms, was adapted for Scopus and Web of Science database searches. The three databases were searched in May 2019 using the developed search strategies (see Appendix A) and the search was updated in March 2020. Key concepts used for searching were “mice”, “blood sample”, “welfare” and “blood sample quality”. Reference lists of all studies selected for critical appraisal were screened for additional studies. Contact with study authors was undertaken where necessary to clarify findings or seek further information. Studies published from database inception were eligible for inclusion.

### 2.3. Study Selection

Following the search, all identified citations were collated and uploaded into EndNote X8.0.1 and duplicates removed. Titles and abstracts were screened by one reviewer (A.W.) for assessment against the inclusion criteria for the review. Potentially relevant studies were retrieved in full and their citation details imported into the Joanna Briggs Institute System for the Unified Management, Assessment and Review of Information (JBI SUMARI, Joanna Briggs Institute, Adelaide, Australia) [15]. The full text of selected citations was assessed in detail against the inclusion criteria by two independent reviewers (A.W. and T.B.). Disagreements that arose between the reviewers at each stage of the study selection process were resolved through discussion. The results of the search, with reasons for study exclusions, are presented in the Preferred Reporting Items for Systematic Reviews and Meta-Analyses (PRISMA) flow diagram (Figure 1) [16].

### 2.4. Assessment of Methodological Quality

Eligible studies were critically appraised for methodological quality by two independent reviewers (A.W. and T.B.) using the SYRCLE risk of bias tool [17]. Any disagreements that arose were resolved through discussion. All studies, regardless of the results of their methodological quality, underwent data extraction and synthesis. Consideration of the methodological quality of individual studies is discussed in the narrative synthesis. 

### 2.5. Data Extraction

Data were extracted from studies included in this review by two independent reviewers (A.W. and T.B.) using an electronic form developed by the authors (see Appendix A for full extraction templates). Extracted data included specific details about the mice populations sampled, the study design, blood sample routes, and the outcomes of significance to the review objective, being an indicator of animal well-being. If the data were presented as figure or other form, they were extracted with the Software Getdata Graph Digitizer 2.26 0.20 (S. Federov, Moscow, Russia). Any disagreements that arose were resolved through discussion. Authors of papers were contacted to request missing or additional data, where required. 

### 2.6. Data Synthesis

Due to the nature of the data extracted, it was decided by the review team that a meta-analysis of any format (including a pairwise meta-analysis or a network meta-analysis) was not appropriate for any data included in this review. In most cases, the studies contributing data towards a particular outcome were extremely heterogeneous. Clinical heterogeneity existed between studies in terms of the intervention timing, frequency of the blood taking procedures, the gauge of the needle used, and the amount of blood taken per procedure. There was also heterogeneity in terms of the characteristics of the mice used (strain, sex, age, etc.). In the few circumstances in which studies were homogeneous enough to facilitate an appropriate meta-analysis, the primary authors rarely provided complete reporting of data, often only reporting *p*-values, a statement of (non-)significance, or simply showcasing their results in the form of a figure or graph that the review team had to “digitize”. The review team are cognizant that ‘digitizing’ data from figures is a subjective, highly variable and imprecise method in which to collect data, and are hesitant to include data collected via this method in any formal meta-analysis.

Because of these limitations and deviations from the methods as specified in the protocol, data were synthesised according to the reporting guidelines of Synthesis Without Meta-analysis (SWiM) [18] for each outcome presented. This has occurred for each outcome and is covered in detail in the results section.

### 2.7. Assessing Certainty in the Findings

The Grading of Recommendations, Assessment, Development and Evaluation (GRADE) approach for grading the certainty of evidence was followed [19,20] and a Summary of Findings (SoF) has been created using the GRADEPro GDT software (McMaster University, ON, Canada) [21]. The SoF reports plasma glucose concentration (mmol/L), plasma corticosterone concentrations (ng/mL), faecal corticosterone concentrations (ng/0.05 gram of faeces) and bodyweight (% change). The SoF has been presented in Table 1.

## 3. Results

### 3.1. Description of Studies

Of the 31 full texts reviewed (Figure 1), 27 articles were eligible for inclusion. The date of publication of these articles ranged from 1995 to 2020, and no relevant studies were identified that were published prior to 1995. Five studies were retrieved through manual searching of the reference list of included studies. The characteristics of the included studies are summarised in Table 2. Observational studies were eligible for inclusion. However, the majority of the studies (25) adopted a randomized controlled study (RCT) design. 

Out of the studies included, 78% evaluated more than one blood sampling method and tended to compare outcomes between blood sample methods. Methods evaluated included: 14 (52%) on facial vein sampling, 14 (52%) on the retrobulbar route, nine (33%) on tail incision, seven (26%) on tail tip amputation, five (19%) on the tail vein method, five (19%) using saphenous sampling, five (19%) on the sublingual, and two (7%) on a non-surgical jugular vein route. A further three studies used miscellaneous methods and uncommon routes of phlebotomy, including the use of blood-sucking bugs, a submental route, and puncture of the tail tip. For reporting, we have used consistent terminology in describing the methods. The definitions we propose are reported below. 

Effects of serial blood sampling were examined in 16 (59%) of studies. Blood sampling interval varied widely across these studies, ranging from a few minutes to 8 week intervals. This finding creates challenges in comparing these studies to examine serial sampling effects. Only four studies (15%) used both male and female mice. Of these studies, 50% reported statistical analysis of sex difference and incorporated findings in data presentation. 

In a number of studies there were associated conditions, which ordinarily might be considered as confounders in study interpretation. These included the use of anaesthesia for sampling, and warming methods for obtaining tail vein samples. Only four out of the 14 studies on the retrobulbar route did not use anaesthesia, whilst 3/5 on the sublingual route were performed conscious. Given that these conditions are regularly used for these methods, and may be mandated by ethics committees, they were considered a part of the method itself and were incorporated in data synthesis. However, where these conditions varied across studies, rendering comparison inappropriate, this has been reported. 

Sample quality measures were reported in six (22%) of the studies. Sample quality was not a primary focus of this review and consequently it should be noted that we utilised a restricted definition of quality, mainly focusing on sample haemolysis and clotting. Furthermore, our search was restricted to studies which looked at quality in conjunction with animal welfare outcomes. We may therefore have not identified all published studies evaluating quality of samples via the different routes.

### 3.2. Note on Terminology

A variety of sampling routes from the tail were described with little consistency in naming. For the purposes of comparison, we have defined as follows: (1) tail amputation involves the removal of the tip of the tail with a blade, (2) tail incision uses a blade to cut the tail laterally, and (3) tail vein is the targeted collection of blood by insertion of a needle directly into the lateral tail vein. 

The submandibular vein, as described in [5], targets the vascular bundle in the caudal part of the jaw. This terminology is commonly used, but perhaps erroneously [39]. The preferred term based on an examination of mouse anatomy is the facial vein [39]. This is the term used in this review, in spite of some usage of ‘submandibular’ in reviewed studies. 

During retrobulbar bleeding, also called retroorbital bleeding, a capillary tube is used to disrupt the retrobulbar venous sinus located behind the eye [8]. Some authors refer to this route as the retroorbital plexus, inferring that a plexus is present in the mouse. This anatomical nomenclature may also be incorrect, although there exists controversy in the claim [39]. We have applied the term ‘retrobulbar’ in summarising these studies. 

The terms saphenous vein, referring to a site near the ankle [8], sublingual for beneath the tongue [4], and jugular, for accessing the vein in the craniocervical region [8] were universally referred to in the studies evaluating them, and are reported as such. 

Finally, the submental vein referred to in one study [33] has also been described as a misnomer by an eminent veterinary anatomist, with a suggestion that the site actually targeted was the inferior labial vein [40]. In spite of this, given that only one study reported on this technique, we have continued to refer to this as the submental route. 

### 3.3. Animal Welfare Outcomes 

Animal welfare is an umbrella term, defined by the summation of the individual summed experiences of an individual [41]. The nature of affective states experienced by the animal, and their relative weighting over time, typically defines whether an animal has, on balance, good or poor welfare [42,43]. Affective states comprise emotions such as pain, fear and joy [44]. For laboratory animals, the term “cumulative experience” has been coined [45]. Cumulative experience has been defined as: ‘the sum of all the events and effects, including their quantity, intensity, recovery between and memory thereof, that impact adversely, positively, and by way of amelioration on the welfare of an animal over its lifetime’ [45]. Whilst phlebotomy techniques are generally considered to produce a short-lived response in an animal, given that they are conducted frequently, they may have a significant impact on cumulative experience that can be minimised through appropriate evidence-based selection of sampling routes. 

The included studies utilised a range of measures for assessing animal affect to provide an indication as to potential welfare impact. Utilising a combination of physiological and behavioural methods is generally regarded as superior in the holistic measurement of welfare state [46,47]. A short description of the main outcomes considered in this review follows. These measures have been categorised into (1) measures of physiology, (2) clinical and pathological parameters and (3) behavioural measures and (4) blood sample quality measures. 

### 3.4. Physiological Measures

The major physiological measures investigated were those reflecting fear or arousal, via the hypothalamic–pituitary–adrenal (HPA) axis response [2,7,8,9,10,11,13,24,31,35,36,38] and the associated release of stored glucose [1,22,23,25,29,32,34,37]. Stress almost invariably activates the HPA axis, which, via a sequence of steps, leads to glucocorticoid production—the principle rodent glucocorticoid being corticosterone. Corticosterone serves to regulate glucose, allowing for its release from reserves in the short term, and carrying out other metabolic actions with the goal of establishing homeostasis [48]. Typically, blood serum or plasma is used to measure corticosterone [49] or glucose concentrations [50], with an increase implying an acute stress response. Measurement of faecal corticosterone metabolites (FCM) has recently been proposed due to the advantage of being non-invasive. FCM provide a retrospective measure of the HPA axis response with lag time from peak in blood to faecal excretion being 9 h in mice [51], thus reflecting sampling method and subsequent recovery [2].

### 3.5. Clinical and Pathological Parameters

General health parameters such as fur condition, inactivity or dehydration status are commonly implemented in severity score sheets for rodent experiments. A change in these parameters can provide an indication of reduced welfare or disease but may not on their own be a sensitive indicator of a stress response. Studies evaluating welfare typically combine these measures with other physiological or behavioural parameters, as occurred in the included studies. Procedure-specific adverse events were important to include as outcome measures, since these may have considerable impact on individual welfare. Furthermore, an increased incidence of such events would prevent recommendation of a technique for practice. Events considered in the studies included haemorrhage from the ear and nares, ocular lesions, circling and convulsions. 

Mortality rate is a commonly used indicator providing a retrospective measure of welfare, since it may be influenced by disease, trauma or environmental problems [52]. 

Other measures utilised are arguably less indicative of animal affective state, but may provide insight by inference—for example, the quantification of tissue damage through post-mortem or histology [1,4,8,24,27,28,29,32,33,36,38] may imply pain or loss of function. 

### 3.6. Behavioural Measures 

Behavioural outcomes were widely reported and ranged from evaluation of spontaneous behaviour, such as eating (ascertained through bodyweight), vocalisation or inactivity, through to the use of well-established behavioural tests of anxiety or affective state [2,8,12,30,32].

Bodyweight loss can result from a variety of causes all of relevance to well-being. These include disease, poor or lack of nutrition, as well as eating behaviour which is potentially compromised by a stress response or trauma associated with a procedure. However, stable bodyweight does not necessarily imply that well-being is not impaired, or even positive in nature [53]. Whilst the use of bodyweight alone as a measure of welfare is fairly crude and non-specific, it is a commonly used surrogate parameter for welfare [54].

Nest building is a spontaneous behaviour that has been proposed to represent a ‘luxury’ behaviour which is highly motivated but non-essential in the laboratory [55]. As such these behaviours are generally the first to be reduced during times of stress [44]. Therefore, poor or reduced performance in this behaviour may indicate a reduction in well-being [55].

Elevated plus maze is used as an assay for anxiety-related behaviour, and typically utilises several different outcome measures to ascertain level of anxiety behaviour, with the general presumption being that increased open arm activity is anti-anxiety behaviour [56]. Open arm activity can be measured as the number of entries or duration. A range of other parameters are frequently collected in this test, including velocity in open and closed arms and distance covered. These are, however, typically a measure of locomotor activity rather than anxiety per se [57]. 

The open field test is used to gather information on ambulation and emotionality [58]. Ambulation or activity can be measured using total distance travelled in the test. Thigmotaxis is used as a measure of anxiogenic behaviour, with thigmotaxis increasing as anxiety increases. This is typically measured through entries into the central zone or time spent in the centre versus the periphery [58].

### 3.7. Blood Sample Quality Measures

Based on our restricted definition of measures of blood sample quality, measures of hemolysis and clotting were considered in our synthesis. Hemolysis is the most common pre-analytical sources of error in clinical laboratory and generally leads to sample rejection and the need for blood re-draw. The finding therefore has animal welfare as well as experimental implications. Furthermore, invisible hemolysis can lead to discharge of cell constituents and false results [59]. Clotting may occur where the blood is slow to fill the collection container, or when considerable manipulation of the vein by the needle has occurred. The presence of clot can therefore give a good indication of the ease with which sample can be collected via the particular route but may also be lessened by operator experience [60]. These samples are unable to be analysed for most laboratory tests.

## 4. Methodological Quality

Overall, the risk of bias of the included studies was high. Often, the nature of the intervention precluded adequate blinding of the operators involved in the blood taking procedures. Whilst it was discussed by the review team that this would be inherent in the included studies, it would still warrant a rating of a “high” for this particular domain. Table 3 details the risk of bias assessments for each domain for included studies, whilst Figure 2 displays the percentage of studies that achieved either a low, unclear or high risk of bias for each domain. See Appendix A for full reviewer judgment for the assessment of methodological quality.

### 4.1. Selection Bias

There was a high risk of selection bias for the majority of the included studies. Only one study was at a low risk of selection bias for all three signalling questions used [27]. The generation of an adequate randomisation sequence was reported in only 8 studies [12,13,22,23,25,27,30,34], while 10 studies [1,2,4,8,11,28,29,32,36,38] stated that randomisation took place but failed to report the actual methods used, thus being assigned as “unclear”. For the remaining studies, randomisation did not take place, or was not mentioned. Baseline characteristics were identified to be similar between groups for only 10 included studies [1,2,8,12,22,23,27,30,31,38], and only 3 [2,27,38] studies reported adequate methodological details on the use of appropriate and proper allocation concealment. 

### 4.2. Performance Bias

There was a high risk of performance bias across all included studies. Due to the design of the studies reviewed and the nature of the blood sampling interventions employed, blinding was not achieved, nor possible. In addition, only seven included studies [12,13,22,25,30,32,38] employed a method of random housing, or utilised a methodology which reduced the possibility of cage associated biases.

### 4.3. Detection Bias

Only 7 [7,10,13,22,25,34,37] of the 27 included studies were recorded as having a “low” risk for both signalling questions used to ascertain detection bias. Whilst blinding of the outcome assessor was not necessarily confirmed for these studies, the majority have been considered to be at low risk as they utilised objective, biochemically validated outcome measures. Of the studies that were of high risk due to the lack of blinding of the outcome assessor, it was mainly due to the inclusion of outcomes such as behavioural assessment that required subjective judgment on the part of the outcome assessor. These were considered to have been at a high risk.

### 4.4. Attrition Bias

Overall, there was a low risk of attrition bias for the included studies. Only four [9,23,24,25] studies were at high risk of attrition bias. One study did not analyse every animal data point [25], inappropriately removing outliers from the reported analysis. Two studies [23,24] experienced loss to follow-up due to failures in the blood collection experimental design. These data were not adequately discussed or analysed in the reported results. Finally, [9] failed to report the number of animals randomised to each group, and therefore, loss to follow-up could not be appropriately assessed. Additionally, four studies [7,10,33,35] were reported as having an unclear risk of attrition bias due to a lack of consistent reporting of animal numbers.

### 4.5. Reporting Bias

Only two studies [26,37] were at high risk of reporting bias. In one of these studies, behavioural observations were stated to be an outcome of interest [26]. However, no results on this outcome have been reported in relation to the bleeding technique. Meanwhile, the reporting in another study [37] was considered to be poor quality, with animal numbers not provided for each treatment group, making assessment of reporting bias difficult.

### 4.6. Other Bias

The only other potential source of bias was the non-disclosure of a funding or supporting body, or lack of conflict of interest. Fourteen included studies were at a high risk of bias due to this factor [1,4,8,13,22,23,24,27,31,32,33,34,35,37]. One study was also considered to be at a high risk of an additional bias, as it was a retrospective review of records from a separately reported, 3 year study [26].

## 5. Effects of the Interventions

Seven outcomes arising from the included studies were synthesised using vote counting based on direction of effect. These outcomes were plasma glucose (single/serial sample), plasma corticosterone (single/serial sample), facial corticosterone, bodyweight change (single/serial sample), nest building hemolysis and clotting. A summary of the results of this synthesis is provided in the Summary of Findings table (Table 1). The full analysis and reporting of these outcomes are provided in Appendix A. Remaining outcomes investigated in the studies are presented narratively below. 

### 5.1. Mortality

Mortality, expressed as a% of study sample based on conversion from the absolute figures, was presented in two studies [26,29]. Both of these studies examined the facial vein route, with [29] additionally comparing with the retrobulbar sample route. These studies evaluated serial sampling at intervals of up to one week. Frolich et al. 2018 also made comparisons with single-sample groups. It is worth noting that the study by Forbes et al. 2010 [26] was a retrospective case-control study with some variation in blood sampling interval and a concurrent study design, potentially confounding interpretation of findings. Due to a sole pairwise comparison both single and serial sampling comparison have been synthesised narratively and are not supplemented with a table.

In a retrospective study [26] with facial vein sampling performed serially, a mortality rate of 4/214 mice was observed (≈2%). Frolich et al. 2018 [29] reported a substantially higher mortality rate of (4/12, 33%) when mice were serially sampled by the facial vein route. There was no associated mortality with the single-sample facial vein route or retrobulbar routes, or serial retrobulbar sampling [29]. Both studies utilised a similar sampling interval of approximately 1 week. 

### 5.2. Adverse Events

Clinical signs or adverse events were considered in four studies [27,29,30,33]. These covered all three of the large sample methods, that of retrobulbar, sublingual and facial vein sample. Regan et al. 2016 also studied submental sampling [33]. There was considerable heterogeneity in the types of adverse events reported which ranged from numbers of repeat attempts at sampling to instances of haemorrage from the site. Events also ranged in severity from mild, such as corneal opacity, to life-threatening. This finding implied that simple addition of incidences of event would provide a biased picture. Furthermore, some adverse events were clearly specific to the sample location, for example ocular lesions or ear canal hemorrhage rendering direct comparison non-meaningful. For this reason, results have been summarised narratively. 

The number of punctures needed to obtain a sufficient blood sample was significantly less for retrobulbar bleeding (1.03 punctures), compared to facial vein (1.45) and sublingual (1.31), (*p* < 0.001 in all comparisons) [30]. Sublingual puncture caused haemorrhage from the nares in 3.33% of mice. Interestingly this rate was increased after the use of anaesthesia to 10.56% of mice [30].

Clinical signs following facial vein bleeding were evaluated in three studies [27,29,30]. Signs included inactivity after collection and being unsteady on release. These occurred at a frequency of 1/20 animals (5%) [27]. The more serious adverse effect of haemorrhage from the ear canal occurred at a rate of 2/20 (10%) [27]. These adverse events occurred when using needle, rather than lancet puncture [26]. Frolich et al. 2018 [29] similarly reported inactivity, ear and nose haemorrhage, as well as head tilt, convulsions, circling and corneal opacity, at rates of approximately 17–25%, but only with serial facial vein samples. Rates of haemorrhage from the ear canal were 2.78%, with an approximate doubling when anaesthesia was used (5.56%) in the Gjendal et al. 2020 study [30].

The most common adverse event reported with retrobulbar sampling was corneal opacity and periocular tissue prolapse occurring at a rate of 2/12 animals (17%) [29]. The incidence of corneal opacity was increased in serial RBB to 5/12 (42%) animals [29]. However, no ocular abnormalities were observed after use of this sampling route in the Gjendal et al. 2020 study [30].

In contrast to other studies Regan et al. 2016 [33], reported no adverse signs after retrobulbar, facial or submental bleeding in a cohort of 15 per group. Minor inflammation was noted at the point of capillary tube insertion in the retrobulbar group but this resolved quickly. 

### 5.3. Histology

Nine studies evaluated histological findings [1,4,8,24,28,29,32,36,38]. Only three of these studies evaluated small vein sample routes [24,32,36]. There was significant heterogeneity in method of reporting which ranged from narrative summary [8,24,32,38], to incidence [1,4,29,36], to a semi-subjective scoring system [28]. There were further differences since lesions were observed in different anatomic regions as would be expected based on sampling location. For example, ocular trauma was reported in retrobulbar sampling, yet was not seen after facial sampling. Similarly, foreign body steatitis is typically caused by hair shaft penetrance of the area and is less likely to occur when sampling hairless regions. In assimilating findings a judgement call has to be made as to whether the increased incidence of a histological finding implies greater welfare impact, or whether this is constituted by greater severity of lesion, or combination thereof. For these reasons, it was considered that vote counting was inappropriate and results have been summarised narratively (Figure 3). 

Retrobulbar sampling consistently led to microscopic evidence of haemorrhage [4,8,28,36] in structures around and within the orbit, including the muscle, retroorbital sinus, harderian gland and nasolacrimal duct. Inflammatory infiltrate with constitute cells changing over time post-sample was also a key feature [4,8,28,36]. Occasional broken hair shafts setting up a foreign body reaction were reported [28,29,36,38]. Optic nerve damage appears to be a rare finding [4]. Massive necrosis, mononuclear cell infiltration, and fibroplasia of the harderian gland was observed in 2/18 animals [36]. 

Jugular vein sampling led to histological changes such as haemorrhage, inflammatory infiltrate, degenerative change, and oedema in muscle, subcutaneous connective and adipose tissue [8].

In facial-punctured animals, macroscopic observations were characterized by subcutaneous haemorrhage and oedema with an acute inflammatory response [1,36]. Focal muscle necrosis was also observed [1]. Signs of trauma persisted for five days [1]. When compared together, sublingual sampling led to fewer traumatic lesions than facial [1]. Lesions in the former were characterized by minimal to slight haemorrhage, with minimal acute inflammation [1,36]. Scar formation and production of granulation tissue was occurring after 5 days [1]. Trichogranuloma has been observed with both of these methods [2,36].

Anaesthesia appears to impact on severity and type of lesions noted. Comparatively fewer indicators of histological change were observed in retrobulbar sampling with anaesthesia compared to the conscious method [8]. In facial vein-sampled animals a diffuse acute neutrophilic/fibrinoid inflammatory response was noted after conscious sampling, whereas the inflammatory response was more chronic in anaesthetized animals [1].This difference in response was likely related to the presence of hair fragments deep within the puncture site in a number of animals [1].

Histology findings following tail sampling were generally mild. Mild neutrophilic inflammation was a consistent finding in mice sampled by tail incision [24,36], which also extended to the dermis in some animals [32]. In a proportion (3/5) of animals, tail amputation resulted in transection of the last caudal vertebra [32]. Additionally all of these animals had neutrophilic inflammation (generally mild) and fibrin at the tail tip [32]. In contrast to tail tip amputation, tail incision led to a shorter period of epidermal oedema, tail muscle involvement with necrosis and inflammatory infiltrate, and earlier proliferation of fibroblasts [36]. The inflammatory infiltrate progressed from neutrophilic to mononuclear cell in both groups over time [36]. Lesions following tail vein sample were primarily in the subcutaneous tissue and adipose tissue [8].

The incidence of histological change was higher after saphenous sample than tail vein (29.4% vs. 13.97% after 1 h) and lesions in the muscle were reported [8]. Alternately, minimal histological change, characterised by minor inflammatory infiltrate and bleeding into the muscle, was observed after saphenous puncture in another study [36]. The authors commented that this finding may have resulted from imperfect tissue sampling. 

### 5.4. Behavioural Tests of Anxiety

#### 5.4.1. Elevated Plus Maze (EPM)

Three studies evaluated EPM performance after blood sampling [2,12,32]. Harikrishnan et al. 2018 [12] investigated all three major blood sampling routes, as well as tail incision. Whilst Moore et al. 2017 [32] examined tail amputation and tail incision in comparison with facial vein sampling. Teilmann et al. 2014 [2] compared facial and tail vein sampling using an EPM as part of a triple test. A further consideration is that the study by Harikrishnan et al. 2018 [12] utilised anaesthesia for retrobulbar sampling, but all other routes were performed conscious in this study and in other studies. This factor may confound study interpretation in spite of the behavioural testing occurring 24 h after sampling. As a result of the differences in experimental design, vote counting was considered inappropriate and results have been summarised narratively. A summary findings table (Table 4) for both the EPM and OFT behavioural test is provided below. 

In Harikrishnan et al. 2018 [12], the groups spent significantly different durations in the open arms (*p* = 0.03), yet there were no differences in time spent in closed arms or the number of open-arm visits. The groups also differed in the number of centre visits (*p* = 0.002). Facially vein-punctured animals did not differ from controls in these parameters, whereas retrobubular sampling caused greatest deviation from controls with reduced centre visits, least time in the centre and reduced activity. Sublingual and tail incision led to intermediate deviations from control values, frequently exhibiting more anxiety behaviour than facially sampled animals, but not differing from each other. 

Moore et al. 2017 examined facial, tail amputation and tail incision routes to find that phlebotomy group did not affect performance in the elevated plus maze [32].

#### 5.4.2. Open Field Test (OFT)

Four studies evaluated OFT performance after blood sampling [2,8,12,32]. Harikrishnan et al. 2018 [12] investigated all three major blood sampling routes, as well as tail incision. Whilst Moore et al. 2017 [32] evaluated tail amputation and tail incision, in comparison with facial vein sampling. Teilmann et al. 2014 [2] compared facial and tail vein sampling using an OFT as part of a triple test. Tsai et al. 2015 [8] studied retrobulbar (with and without anaesthesia), facial, tail vein, and saphenous bleeding.

Tsai et al. 2015 [8] measured total distance travelled in the test which is an important measure of locomotor activity. Distance travelled differed between groups being longest in mice that underwent tail vein bleeding (2.183 cm) or saphenous (2.110 cm), followed by mice that underwent retrobulbar (1.699 cm) and facial vein bleeding (1.226 cm). This difference was significant for facially vein-sampled animals, but non-significant for other pairwise comparisons. Facial vein-sampled animals also had a lower average speed. This study has not been included in the summary table because values were not compared to a sham/unmanipulated control to enable a determination on direction of effect. 

The study by Harikrishnan et al. 2018 [12] utilised anaesthesia for retrobulbar sampling but all other routes were performed conscious in this and in other studies. This may be a consideration in interpretation, in spite of testing occurring 24 h after sampling. 

Mice subjected to retrobulbar sinus puncture were significantly less active than control mice for all three OFT parameters in Harikrishan et al. 2018 [12]. This was not observed for mice that underwent tail incision or facial vein puncture. An effect of anaesthesia was apparent since mice subjected only to isoflurane anaesthesia showed greater activity, and a higher number of centre entries, than mice subject to retrobulbar puncture under isoflurane anaesthesia [12]. This contrasts with the Tsai et al. 2015 study [8], where mice that underwent retrobulbar bleeding with anaesthesia performed similarly in the OFT to mice sampled without anaesthesia. 

There were no differences in display of anxiety behaviour between facial vein-sampled, tail incision, sublingual and retrobulbar groups, as indicated by visits and time spent in the centre of the field [12]. Alternately, mice sampled by facial vein generally avoided the open field in the triple test with the authors concluding that they expressed more anxious behaviour than tail vein-sampled animals [2].

In Moore et al. 2017 [32], the facial vein group exhibited a significantly lower average speed in the OFT compared to tail amputation and incision groups. No other between-group differences were observed.

## 6. Discussion

As common techniques performed in biomedical research studies involving mice, blood sampling methods may have considerable impact on cumulative experience. Furthermore, retrobulbar sampling has been considered controversial and its use is vetoed in some laboratories. Whilst summary documents or guidelines exist on this topic, a number of these are now older documents and it is not clear whether they have been based on a systematic review of the available evidence (see, e.g., [3,61]). This review represents the first systematic review on recovery blood sampling techniques in mice based on their impact on animal welfare and sample quality. 

### 6.1. Impact of Blood Sample Route on Mouse Welfare

Whilst there is a substantial body of evidence on the impact of blood sampling on animal welfare with 27 studies sourced, the heterogeneity in terms of sampling routes compared, and outcomes measured, renders it problematic to make any recommendation. Despite the large number of studies, few performed the same pairwise comparisons, with the same outcome measures. We were also unable to perform any assimilation of the behavioural data given the heterogeneity. This is unfortunate, as arguably these measures may provide a better measure of well-being and animal impact than a measure of the short-term stress response. Given the above caveat, some general points taken from the assimilation are presented. 

For serial blood sampling, as a general rule, small-volume sampling routes may be beneficial over large-volume sampling routes based on the findings for glucose, plasma corticosterone and bodyweight (Table 1). This would seem plausible based on physiological principles given the reduced blood volume lost, and therefore reduced chance of hypotension, haemorrhagic shock and impaired tissue and organ metabolism that can result [62].

Whilst it might be expected that head-focused routes would have a greater impact on bodyweight than methods focused at the tail and leg, this was only borne out for the facial and retrobulbar routes. The finding that bodyweight loss did not occur with sublingual sampling (data from two studies) is intriguing given the mouth focus, and requires further study to confirm. 

It is often assumed that anaesthesia improves animal well-being following procedure performance, yet the findings are inconsistent. For example, beneficial effects of isoflurane use were found, with reduced anxiety being observed in the OFT [12] and decreased histological lesion severity after retrobulbar and facial sampling [1,8]. However, anaesthesia caused a doubling in the incidence of the serious adverse effect of ear bleeding after facial vein bleeding, and a tripling in rate of bleeding from the nares after sublingual puncture as observed in [30]. Perhaps the best current advice in this regard is to determine anaesthetic use on a case-by-case basis, dependent on the blood sampling route, and the researcher’s level of technical expertise and comfort with the procedure. These apparent discrepancies also need future targeted research focus. 

The difference in serial mortality rate observed for the facial vein route between the studies of [26,29] (2% vs. 33%) is striking. This is especially so since the technique is widely used in laboratory animal practice, and this mortality rate would likely raise ethical concerns for continued performance amongst many ethics committees. These results may be an artefact of the small sample sizes employed in the Frolich et al. 2018 study, with a potential effect of learning/re-familiarisation. Mortality rates may have declined as the skill was reacquired with larger numbers of animals sampled. Only one of the included studies specifically investigated the effect of experience on outcomes measured [28]. Based on pathological findings, findings from this study were that experience with retrobulbar sampling had little impact on outcome. However, it is suggested that this effect may have been overlooked in the included studies, and deserves further research attention, and an agreed criterion to standardise expertise level. 

As a final point, based on the evidence available, the reason for the demise of the retrobulbar route for ethical reasons remains unclear. The synthesis implies that it is associated with zero mortality [29], none [30,33] to mild clinical ocular abnormalities [29], and similar severity of histological lesions to other large-volume sampling routes [36,38]. This change in policy direction is more surprising given that this route has generally been replaced by facial vein sampling, which arguably can lead to a similar number, if not more, potentially negative outcomes [27,29,33,38]. It is speculated that pictures, or dialogue showing the horrendous (likely rare) outcome of globe perforation after sampling, may have influenced decision making in this regard. Perhaps a more appropriate focus for policy makers should be how to best train and ensure operator competency in this technique in order to avoid the occurrence of serious adverse effects. It is certainly by no means clear whether the proposed alternative facial route is more beneficial for animal well-being and this should be a priority for future research. 

### 6.2. Evidence Completeness and Quality and Recommendations for Future Research 

This review identifies a number of factors preventing recommendations on choice of mouse blood sampling route being made. These include (1) that in spite of a reasonable number of studies on the topic, there were often few studies examining the same pairwise comparison; (2) there was a lack of standardised outcome measures relevant to well-being and timepoints for comparison; (3) many studies were at high risk of bias, either by virtue of study design or deficiencies in reporting. When considering that there have been significant and repeated recent efforts to improve the reporting standards of animal research, and that guidelines to assist animal research have been widely available since 2010 [63,64,65,66], the overall poor quality of reporting of the included studies is problematic. Simple details prescribed by the ARRIVE (Animal Research: Reporting of In Vivo Experiments) guidelines, such as description of the randomisation procedure, were only reported by three studies published after 2010 (of 16 total). This highlights how far animal-based research needs to come to be considered rigorous, transparent and transferable.

Generally, RCTs are desirable due to their placement in the hierarchy of evidence. However, given the practical and widespread nature of these interventions, and therefore the importance of external validity, it may be desirable to investigate these techniques using large-scale, well-designed observational studies. This may avoid issues alluded to earlier, where small sample sizes may lead to artefacts. Alternatively, RCTs that are multicentre in nature to increase sample size could be performed. These studies may be able to reduce confounding by issues such as technique experience as part of a dilution effect due to the use of many operators.

One of the major limitations of this systematic review was the inability to perform a meta-analysis, which is the ideal method to synthesise and present data from multiple, comparable studies. However, due to the heterogeneity that existed between studies, performing a meta-analysis was never considered to be appropriate. While this factor has severely limited the impact of our results in terms of making a determination on the effects of blood sample route on animal welfare, it does provide us some insight as to how a meta-analysis may be facilitated for future systematic reviews on this topic. Ideally, the relevance and reproducibility of outcome measures used to assess welfare in adult mice should be validated and discussed in the field, so that consensus may be reached, and experiments standardised accordingly. However, by far the most common barrier to performing a meta-analysis encountered was the underreporting of results by authors of the primary studies. Often, authors simply reported their results as a figure or a graph, and a statement of (non-)significance. While this may suit the purposes of the primary author in confirming or rejecting their null hypothesis, this underreporting greatly reduces the transferability and comparability of these data. A simple solution to this problem would be for scientific journals to start mandating that submitting authors provide complete data sets. This tactic is already being employed by international journals such as PLoS One, Springer Nature and Science [67,68,69] and may reduce the need to resort to alternate, less robust data synthesis strategies [70]. However, until this becomes a standard, as it has in clinical research, this issue will continue to plague animal-based research.

## 7. Conclusions

In spite of a substantial body of evidence investigating welfare associated with blood sampling techniques in mice, it was concluded that there was not enough, high-quality evidence to make any recommendations on the optimal method of blood sampling from the point of view of animal welfare. Future high-quality studies, with standardised outcome measures and large sample sizes, are required.

There is an urgent need, as highlighted by many authorities, to increase quality (and/or reporting) of animal research at all stages from inception to reporting. The use of guidelines such as those published by ARRIVE [64], and protocol registration, can assist in achieving this. Journal editors also need to advise researchers of guidelines and enforce provisions, which will no doubt serve as an educative as well as compliance function.

## Figures and Tables

**Figure 1 animals-10-00989-f001:**
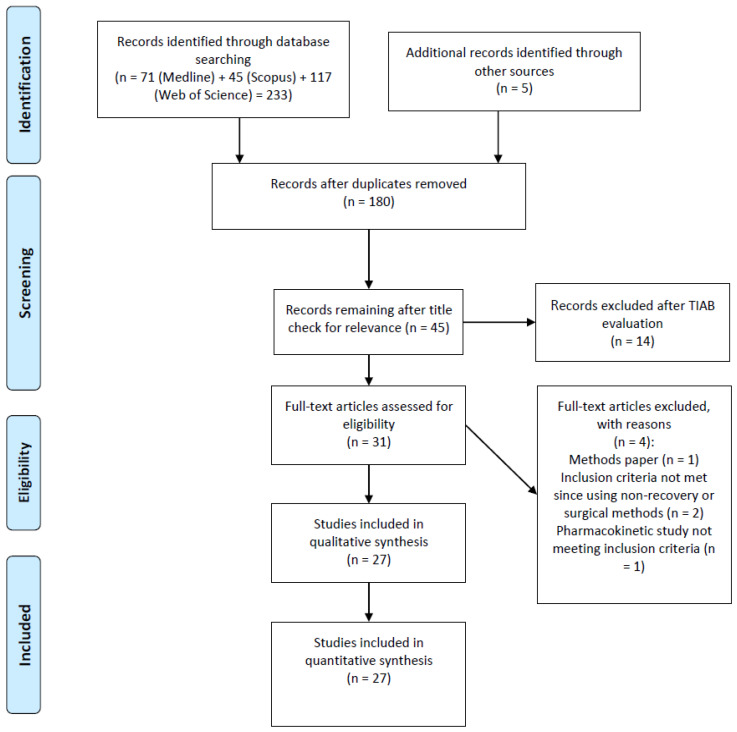
The PRISMA flow diagram for the systematic review detailing the database searches, the number of abstracts screened and the full texts retrieved.

**Figure 2 animals-10-00989-f002:**
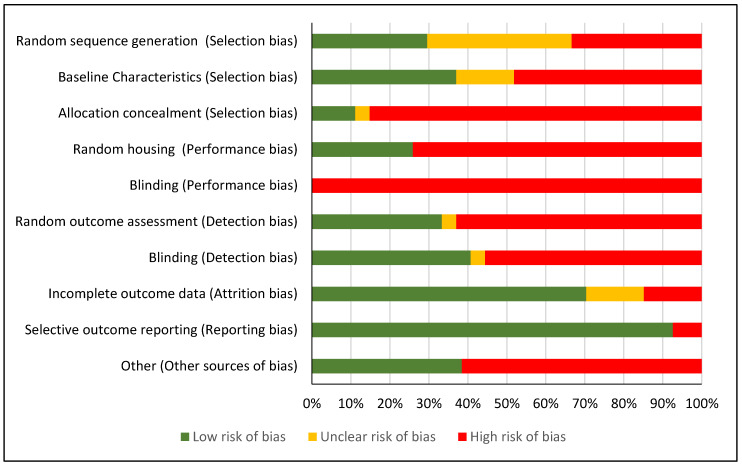
Percentage of studies that achieved either a low, unclear or high risk of bias for each domain using the SYstematic Review Centre for Laboratory animal Experimentation (SYRCLE) risk of bias tool. Although randomisation was mentioned in several articles, lack of reporting of the method used resulted in an unclear risk of bias for most items. Blinding was impossible to achieve due to the inherent nature of the study design.

**Figure 3 animals-10-00989-f003:**
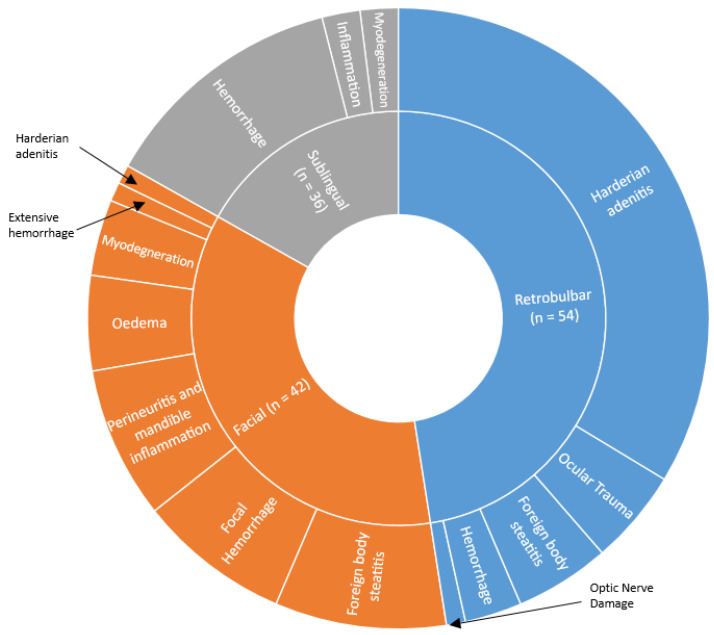
Histological characteristics observed in studies comparing large-volume sample sites. Size of slice represents relative incidence. Difference in reporting prevented assimilation of all findings. Compiled from [1,4,29,32,38]. Note that animals may have demonstrated more than one histological finding. *n* represents the number of animals.

**Table 1 animals-10-00989-t001:** Grading of Recommendations, Assessment, Development and Evaluation (GRADE) Summary of Findings table for primary outcomes.

The Effects of Various Phlebotomy Techniques for Animals Welfare-Related Outcomes			
Patient or Population: Laboratory Mice Used for Research Purposes Interventions of Interest: Sublingual Sampling, Retrobulbar Sampling, Facial Vein Sampling, Tail Amputation Sampling, Tail Incision Sampling, Tail Vein Sampling and Saphenous Vein Sampling			
Outcomes	Impact	Number of Participants (Studies)	Certainty of the Evidenc (GRADE)
**Plasma Glucose Concentration**			
Single Sampling	The facial vein technique was involved in the most pairwise comparisons synthesised, and was found to have a beneficial effect when compared to the tail amputation and the tail incision techniques. The tail incision technique was likewise considered to be beneficial compared to the tail amputation method. The retrobulbar technique was considered beneficial to the tail vein technique, and the sublingual technique beneficial to the facial vein technique.	(3 RCTs)	⨁◯◯◯ VERY LOW ^a,b,c^
Serial Sampling	The retrobulbar technique was involved in 4 pairwise comparisons and was considered harmful compared to the tail amputation and the tail incision methods. Yet it was beneficial when compared to the facial vein method. Tail amputation was similarly beneficial over the tail incision technique, and the saphenous technique was beneficial compared to the tail vein technique.	(3 RCTs)	⨁◯◯◯ VERY LOW ^a,c,d^
**Plasma Corticosterone Concentration**			
Single Sampling	The retrobulbar technique proved to be more beneficial compared to the facial vein technique in two studies, beneficial to the tail vein and saphenous methods in one study each, yet harmful when compared to the tail incision technique based on the results from one study. Neither a harmful or beneficial effect was observed comparing the facial vein and the tail vein method; the tail amputation and the saphenous technique; or the tail vein and the tail incision method.	(5 RCTs)	⨁◯◯◯ VERY LOW ^a,e,f^
Serial Sampling	The tail amputation technique was considered beneficial in two pairwise comparisons against the retrobulbar technique, and beneficial in a pairwise comparison against the facial vein technique. However, no difference was observed when compared with the saphenous vein technique. Likewise, the saphenous vein technique was considered beneficial when compared against both the retrobulbar and the facial vein technique.	(3 RCTs)	⨁◯◯◯ VERY LOW ^a,e,g^
**Faecal Corticosterone**			
Serial Sampling	There appears to be a time-dependent factor for the pairwise comparison of the retrobulbar technique and the tail incision technique. While the tail vein was shown to be beneficial compared to the facial vein.	(4 RCTs)	⨁◯◯◯ VERY LOW ^a,h,i^
**Body Weight**			
Single Sampling	The saphenous vein technique was considered ‘beneficial’ in pairwise comparisons with the sublingual, retrobulbar, facial, tail amputation and tail incision methods. This technique was not considered ‘harmful’ in any pairwise comparison synthesised. The next most beneficial technique synthesised was the sublingual technique, which was considered beneficial in two separate comparisons made to the facial vein technique, and single comparisons made to the retrobulbar; tail amputation and tail incision techniques. The tail amputation technique was beneficial over the retrobulbar; facial vein and the tail incision methods. The retrobulbar and facial vein methods were associated with the most harm, as compared through pairwise comparison.	(5 RCTs)	⨁◯◯◯ VERY LOW ^a,j^
Serial Sampling	The retrobulbar technique was considered beneficial in two separate pairwise comparisons made to the facial vein technique, as well as being beneficial when compared to the sublingual technique.	(3 RCTs)	⨁◯◯◯ VERY LOW ^k,l^
**GRADE Working Group grades of evidence:** High certainty: We are very confident that the true effect lies close to that of the estimate of the effect. Moderate certainty: We are moderately confident in the effect estimate: the true effect is likely to be close to the estimate of the effect, but there is a possibility that it is substantially different. Low certainty: Our confidence in the effect estimate is limited: the true effect may be substantially different from the estimate of the effect. Very low certainty: We have very little confidence in the effect estimate: the true effect is likely to be substantially different from the estimate of effect.			

^a^ Downgraded two levels for risk of bias as there is a high risk of performance bias across all contributing studies. ^b^ Downgraded two levels for inconsistency. The direction and magnitude of pairwise comparisons varied across the different studies. In addition, one comparison involved animals that were anaesthetised ^c^ Downgraded one level for imprecision—only 3 studies of limited size contributed data towards this outcome and no two studies included the same pairwise comparison, disallowing a synthesised pairwise comparison to be obtained. ^d^ Downgraded two levels for inconsistency. The direction and magnitude of pairwise comparisons varied across the different studies. ^e^ Downgraded one level for inconsistency. Of the one pairwise comparison that was presented in two separate studies, the same direction of effect was observed. However, multiple other comparisons included in the outcome were regarded as having no difference. ^f^ Downgraded one level for imprecision. Five adequately powered studies have contributed towards this outcome synthesis and there are multiple studies that provide data towards the same pairwise comparison. However due to the nature of synthesis we have borderline concerns with imprecision. ^g^ Downgraded one level for imprecision—only 3 studies of limited size contributed data towards this outcome and only one pairwise comparison was included across multiple studies ^h^ Downgraded two levels for inconsistency. Few different pairwise comparisons were made across studies, of the one comparison that was synthesised across multiple studies the result differs substantially ^i^ Downgraded one level for imprecision—only 3 small studies have contributed data towards this outcome synthesis ^j^ There are multiple pairwise comparisons made for this outcome and the direction of effect appears to be consistent. However, the review team believed that the domains of both inconsistency and imprecision were borderline calls and have downgraded one level between the two ^k^ Downgraded one level for inconsistency. Only one pairwise comparison was observed over multiple studies. However, the direction of that comparison occurred in the same direction across studies. Wide variation in results from other pairwise comparisons across results from included studies ^l^ Downgraded one level for imprecision—only 3 studies of limited size contributed data towards this outcome. Primary outcome data considered using GRADE are bolded.

**Table 2 animals-10-00989-t002:** Summary of included studies.

Author	Strain	Sex	Age at Intervention	Animals per Group	Study Design	Intervention	Accompanying Conditions	Frequency of Intervention (Occasions)	Comparator	Outcome
Aasland et al. 2010 [22]	C57BL/6JBomTac	M	NR	n = 8	RCT (crossover)	Saphenous vein Tail vein	N/A	4 by both methods at 2 weeks apart	Each other Time series	Plasma glucose Hemolysis
Abatan et al. 2008 [9]	ICR	F	NR	n = 8–13 (unclear from methods)	RCT	Saphenous vein Tail tip amputation	N/A	One-off collection Serial sample at 2–3 day intervals (4)	Each other Time series	Plasma corticosterone Behaviours noted during blood collection
Christensen et al. 2009 [23]	C57BL/6JBom	M	NR	n = 20	RCT	All sample methods were performed at 21 and 30 °C, i.e., 8 experimental groups Retrobulbar Tail incision Tail tip amputation Tail tip puncture	N/A	Serial sample at 30 min intervals (4)	Other groups	Blood glucose Haemolysis Clotting
Durschlag et al. 1996 [24]	ICR	M	9 weeks	n = 8	Case report	Tail incision	N/A	Serial sample at 2–3 day intervals (5)	Time series	Histology Plasma corticosterone
Fernandez et al. 2009 [25]	C57BL/6J	M	6 weeks	n = 10	RCT (crossover)	Retrobulbar Facial vein	Anaesthesia (retrobulbar)	Serial sample at 6–8 week intervals (3)	Each other	Blood glucose Haemolysis
Forbes et al. 2010 [26]	Balb/c	F	6–8 weeks	n = 214	Retrospective case series	Facial vein	Lancet or needle for sampling	Serial sample at 2–7 day intervals (6)	Nil	Mortality
Francisco et al. 2015 [27]	BALB/c	F	5 weeks	n = 20	RCT	Facial vein (lancet or needle)	Facial vein route with anaesthesia as one group	One-off collection	Other groups	Adverse events Gross post-mortem site evaluation
Fried et al. 2015 [28]	C57BL/6N background with a mutation in MDA5	M/F	4–6 months	n = 8	RCT	Retrobulbar	Anaesthesia	One-off collection	Time series at 0, 1, 3, 7 or 14 days after sampling	Clinical scores Histology
Frolich et al. 2018 [29]	C57BL/6NCrl	F	12–14 weeks	n = 12	RCT	Retrobulbar Facial vein	Anaesthesia (retrobulbar)	One-off collection Serial sample at 1 wk intervals (6)	Each otherSingle versus serial	Adverse events Mortality Bodyweight Histology Plasma glucose
Gjendal et al. 2020 [30]	C57BL/6	F	10 weeks	n = 30	RCT	Retrobulbar Facial vein Sublingual	Anaesthesia (retrobulbar) Facial vein and sublingual anaesthesia groups, in addition to conscious sampling	Serial sample at days 8, 9, and 10 (short protocol) and days 8, 15, and 22 (long protocol)	Eachother Single versus serial	Nest build score Faecal corticosteroid metabolites Bodyweight Haemolysis Clotting Gross post-mortem site evaluation
Harikrishnan et al. 2018 [12]	C57BL/6NTac	M/F	6 weeks	n = 12	RCT	Retrobulbar Sublingual Facial vein Tail incision	Anaesthesia (retrobulbar)	Serial sample at 24 h interval (2)	Other groups, plus isoflurane control, and behavioural test control (naïve animals)	Nest build score Elevated plus maze Open field test Faecal corticosteroid metabolites
Heimann et al. 2009 [4]	Crl: CD-1 [CR]	M/F	11 weeks	n = 30	RCT	Sublingual Retrobulbar	Anaesthesia	One-off collection	Each other	Histology
Heimann et al. 2010 [1]	CD1	F	14 weeks	n = 18	RCT	Sublingual Facial vein	Anaesthesia (for sublingual and one facial vein group)	One-off collection	Other groups, time series at 3 h, two or five days after sampling	Bodyweight Food intake Histology Blood glucose
Kim et al. 2018 [10]	CD-1 C57BL/6	M	NR	n = 4–6	RCT	Retrobulbar Tail tip amputation	Anaesthesia (retrobulbar) Restrain/unrestrained	One-off collection Serial sample tail tip groups at 30 min intervals (5)	Other groupsTime series	Plasma corticosterone
Madetoja et al. 2009 [31]	Hsdwin:NMRI	F	9 weeks	n = 10	RCT	Saphenous vein Facial vein Tail vein	Tail warming heat lamp	One-off collection	Other groups, and control with no blood samples	Plasma corticosterone Plasma ACTH
Moore et al. 2017 [32]	C57BL/6J	M	10–12 weeks	n = 8–12	RCT	Facial vein Tail tip amputation Tail incision	N/A	One-off collection	Other groups, and sham submandibular and tail tip amputation (just restraint)	Blood glucose Audible vocalisations Post-procedural epochs of inactivity Grooming behaviour Nest build score Elevated plus maze Open field test Histology
Regan et al. 2016 [33]	CD1	F	12–13 weeks	n = 15	RCT	Retrobulbar Facial vein Submental	Anaesthesia (retrobulbar)	Serial sample at 2 week intervals (3)	Other groups	Adverse events Bodyweight Extraneous blood loss Gross post-mortem site evaluation Haemolysis Clotting
Rogers et al. 1999 [34]	**Study 1** (single sample) C57BL/6	F	10–12 weeks	n = 72	RCT (crossover)	Retrobulbar	Thermostatic warming chamber	One-off collection (each method)	Each other	Plasma glucose
	**Study 2** (repeated sample) C57BL/6	F	10–12 weeks	n = 48		Tail incision		Serial sample at 1 week interval (2)		Plasma insulin
Sadler et al. 2013 [7]	**Study 1** (single sample) BALB/CAnNCrl **Study 2** (repeated sample) BALB/CAnNCrl	M M	7–8 weeks 3–4 weeks	n = 5 n = 4/5	RCT	Tail incision Tail vein Tail incision	Thermostatic warming chamber Tail dipped hot water	One-off collection Serial sample at 24 h intervals (3)	Other groups Time series	Plasma corticosterone
Shirasaki et al. 2012 [35]	ICR C57BL/6N	M	6 weeks	n = 10 (unclear from methods)	RCT	Jugular Tail incision	N/A	One-off collection Serial sample at 24 h intervals (5)	Each other Time series	Plasma CRP Plasma corticosterone Plasma haemoglobin Hematocrit Plasma thrombin–antithrombin complexes
Sorenson et al. 2019 [36]	C57BL/6NTac	F	8 weeks	n = 36	RCT	Retrobulbar Saphenous Sublingual Facial vein Tail incision Tail tip amputation	Anaesthesia (retrobulbar)	One-off collection	Each other, plus isoflurane control and naïve animals (no bleeding) Time series at nine timepoints: 6 or 10 h or 1, 2, 4, 6, 8, 10, or 12 days after sampling	Bodyweight Stomach contents Plasma corticosterone Inflammatory gene expression at sample site Plasma inflammatory markers (haptoglobin and IL1ß) Histology
Tabata et al. 1998 [37]	**Study 1** (single sample) B6C3Fl/ICR **Study 2** (repeated sample) B6C3Fl	M/F M/F	8 weeks 9 weeks	n = 12 n = 20	RCT	Tail tip amputation	Tube restraint or anaesthetized with ether or pentobarbital ^§^	One-off collection Serial sample at varied intervals for 24 h (8)	Other groups Time series	Plasma glucose
Teilmann et al. 2014a [2]	BomTac:NMRI	M	6–8 weeks	n = 8–18	RCT	Facial vein Tail vein	N/A	Two day and two night samples within 24 h (4)	Other groups, plus controls (no blood sample and naïve animals as behavioural control)	Bodyweight Plasma corticosterone Faecal corticosteroid metabolites Triple test (elevated plus maze, open field test, light–dark box)
Teilmann et al. 2014 [38]	C57BL/6J	M	5 months	n = 8–12	RCT	Retrobulbar Facial vein	N/A	One-off collection	Other group, plus control (no blood sample)	Bodyweight Plasma corticosterone Histology
Tsai et al. 2015 [8]	BALB/cO1aHsd	F	8 weeks	n = 12	RCT	Jugular Retrobulbar (with and without anaesthesia) Saphenous Facial vein Tail vein	Anaesthesia (retrobulbar group and jugular)	One-off collection	Other groups	In-cage activity Plasma corticosterone Open field test Histology
Tuli et al. 1995 [11]	**Study 1** (acute stress) BALB/c/Ola **Study 2** (tail bleeding recovery) BALB/c/Ola	M M	5–6 months 5–6 months	n = 5 n = 5	RCT	Tail tip amputation	Tail dipped hot water	One-off collection each route Serial humane killing at day 2, 4 and 8 after blood sample	Other groups and control with no tail amputation	Plasma corticosterone Adrenal weight Spleen weight
Voigt et al. 2013 [13]	C57BL/6CrlN	F	4–6 months	n = 36 (16 contributed to final results due to technical failure)	RCT (crossover)	Blood-sucking bug Retrobulbar Tail incision	Anaesthesia (retrobulbar)	One-off collection each route (note crossover design)	Other groups	Faecal corticosteroid metabolites

NR—not reported. ^§^—The aim of this section of the study was to investigate common scenarios, such as anaesthesia, which elevated blood glucose in laboratory mice, rather than investigate the blood sampling technique per se. The reviewers still considered the study was worthy of inclusion since it could provide data on the topic but have only extracted data relevant to the review question. Note: Restraint not included as an accompanying condition unless effects of this specifically investigated as part of study design, since this would be needed for all sample routes.

**Table 3 animals-10-00989-t003:** Risk of bias assessments for each domain for included studies. L: low risk of bias, U: unclear risk of bias, H: high risk of bias.

Study	Random Sequence Generation	Baseline Characteristics	Allocation Concealment	Random Housing	Blinding	Random Outcome Assessment	Blinding	Incomplete Outcome Data	Selective Outcome Reporting	Other
Aasland et al. 2010 [22]	L	L	H	L	H	L	L	L	L	H
Abatan et al. 2008 [9]	H	H	H	H	H	H	H	H	L	L
Christensen et al. 2009 [23]	L	L	H	H	H	H	L	H	L	H
Durschlag et al. 1996 [24]	H	H	H	H	H	H	H	H	L	H
Fernandez et al. 2009 [25]	L	H	H	L	H	L	L	H	L	L
Forbes et al. 2010 [26]	H	U	H	H	H	H	H	L	H	H
Francisco et al. 2015 [27]	L	L	L	H	H	L	H	L	L	H
Fried et al. 2015 [28]	U	H	H	H	H	H	H	L	L	L
Frolich et al. 2018 [29]	U	H	H	H	H	H	H	L	L	L
Gjendal et al. 2020 [30]	L	L	H	L	H	H	L	L	L	L
Harikishnan et al. 2018 [12]	L	L	H	L	H	L	H	L	L	L
Heimann et al. 2009 [4]	U	H	H	H	H	H	H	L	L	H
Heimann et al. 2010 [1]	U	L	H	H	H	H	H	L	L	H
Kim et al. 2018 [10]	H	U	H	H	H	L	L	U	L	L
Madetoja et al. 2009 [31]	H	L	U	H	H	H	H	L	L	H
Moore et al. 2017 [32]	U	H	H	L	H	H	H	L	L	H
Regan et al. 2016 [33]	H	H	H	H	H	H	H	U	L	H
Rogers et al. 1999 [34]	L	H	H	H	H	L	L	L	L	H
Sadler et al. 2013 [7]	H	U	H	H	H	L	L	U	L	L
Shirasaki et al. 2012 [35]	H	H	H	H	H	H	L	U	L	H
Sorenson et al. 2019 [36]	U	H	H	H	H	H	H	L	L	H
Tabata 1998 [37]	H	H	H	H	H	L	L	L	H	H
Teilmann et al. 2014a [2]	U	L	L	H	H	H	H	L	L	L
Teilmann et al. 2014b [38]	U	L	L	L	H	U	U	L	L	L
Tsai et al. 2015 [8]	U	L	H	H	H	H	H	L	L	H
Tuli et al. 1995 [11]	U	H	H	H	H	H	L	L	L	L
Voigt et al. 2013 [13]	L	U	H	L	H	L	L	L	L	H

**Table 4 animals-10-00989-t004:** Summary of direction of effect for behavioural tests elevated plus maze (EPM) and open field test (OFT) when compared to sham/unmanipulated controls or baseline values.

Single-Sample Method and Reference	Behaviour Test	Measure	General Direction of Effect on Measure	Timeframe for Measure
Retrobulbar				
Harikrishnan et al. 2018 [12]	EPM	Anxiety	↑	24 h
	OFT	Anxiety	↑	24 h
	OFT	Locomotor activity	↓	24 h
Facial Vein				
Harikrishnan et al. 2018 [12]	EPM	Anxiety	=	24 h
	OFT	Anxiety	=	24 h
	OFT	Locomotor activity	=	24 h
Moore et al. 2017 [32]	EPM	Anxiety	=	Few hours post-procedural
	OFT	Anxiety	=	Few hours post-procedural
Teilmann et al. 2014 [2]	Triple Test	Anxiety	↑	24 h
Sublingual				
Harikrishnan et al. 2018 [12]	EPM	Anxiety	↑	24 h
	OFT	Anxiety	=	24 h
	OFT	Locomotor activity	↓	24 h
Tail Incision				
Harikrishnan et al. 2018 [12]	EPM	Anxiety	↑	24 h
	OFT	Anxiety	=	24 h
	OFT	Locomotor activity	=	24 h
Moore et al. 2017 [32]	EPM	Anxiety	=	Few hours post-procedural
	OFT	Anxiety	=	Few hours post-procedural
Tail Vein				
Teilmann et al. 2014 [2]	Triple Test	Anxiety	↓	24 h
Tail Amputation				
Moore et al. 2017 [32]	EPM	Anxiety	=	Few hours post-procedural
	OFT	Anxiety	=	Few hours post-procedural

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
