# Peer review of "The Impact of Common Recovery Blood Sampling Methods, in Mice (Mus Musculus), on Well-Being and Sample Quality: A Systematic Review"

_animals, 2020, doi:10.3390/ani10060989_

Round 1
Reviewer 1 Report
The manuscript by Whittaker and Barker represents a systematic review of the impact of common blood sampling in mice.
The review is very well written and the authors have made a great effort to carry out this study.
The criteria and strategies used are clearly defined and the results are thoroughly discussed.
The amount of data presented and discussed is remarkable, and has an impact on the readability of the several tables in the appendix.
- In my opinion, the Simple summary could be removed because it does not shorten the abstract, so it is useless.
- Pag. 2 Table 1
To improve the readability, I could suggest to put together the lines related to the same outcome (plasma glucose concentration, plasma corticosterone concentration and body weight) dividing them into two sub-rows for single vs. serial sampling.
-Pag. 23 Fig. 3
What does the third arrow in the retrobulbar subunit represent?
- Pag 19 Table 3
Improve the readability. For example the type of bias (selection, performace, detection etc) could be removed here, it has already been clarified in Fig. 2
Author Response
- In my opinion, the Simple summary could be removed because it does not shorten the abstract, so it is useless.
We are happy to remove but will take editorial advisement as this is part of the journal requirements.
- Pag. 2 Table 1
To improve the readability, I could suggest to put together the lines related to the same outcome (plasma glucose concentration, plasma corticosterone concentration and body weight) dividing them into two sub-rows for single vs. serial sampling.
We have amended the Summary of Findings tables according to the peer-reviewer comments.
-Pag. 23 Fig. 3
What does the third arrow in the retrobulbar subunit represent?
This was an issue with the formatting of the image. This has been amended in the updated manuscript
- Pag 19 Table 3
Improve the readability. For example the type of bias (selection, performance, detection etc) could be removed here, it has already been clarified in Fig. 2
We have amended table 3 according to the suggestions made by the reviewer.
Reviewer 2 Report
It is a very interesting and excellent written Systematic Review on the impact of common recovery blood sampling methods, in mice, on wellbeing and the sample quality. This manuscript is coming to fill a gap on this topic. It is also an excellent example of methodology on how to perform a systematic review in the field of Laboratory Animal Science.
At the beginning of the revision it was questionable if it is really necessary to include Appendixes B and C in the manuscript (it is too long). By finishing the revision I was convinced that all Appendixes should be included because the provided information is very useful for the reader.
Three minor issues/comments for authors' consideration:
The first one is related to the chronological period which was covered for search. As papers included in the search were between 1995 and 2020, it would be interesting to show to the readers the range of search. On the contrary it will interesting to state that no relevant papers were found before 1995.
The second comment is related to figure 1 of page 7. There 5 records identified through other sources. Which sources?
Finally a third comment. On page 5 line 119. It is stated that the search strategy aimed to locate both published and unpublished studies. Can the authors clarify their strategy to locate unpublished studies?
Author Response
Thank you for your comments- responses are below.
The first one is related to the chronological period which was covered for search. As papers included in the search were between 1995 and 2020, it would be interesting to show to the readers the range of search. On the contrary it will be interesting to state that no relevant papers were found before 1995.
Thank you for the comment, we have included this information in lines 204 to 206 of the updated manuscript. This section now says that no relevant articles were identified prior to 1995.
The second comment is related to figure 1 of page 7. There 5 records identified through other sources. Which sources?
The records identified through other sources were described from lines 142 to 145 under the ‘Search Strategy’ subheading in the methods. This source was manual reference list checking of included studies which is standard methodological approach for systematic review searching. This has been clarified in the results section at lines 205-206 of the updated manuscript.
Finally a third comment. On page 5 line 119. It is stated that the search strategy aimed to locate both published and unpublished studies. Can the authors clarify their strategy to locate unpublished studies?
The sentence regarding unpublished studies has been removed. Whilst unpublished studies (conference abstracts) were found during the searching of the databases listed in the search strategy, no unpublished source databases were formally searched.